# Quantum Computation and Measurements from an Exotic Space-Time $R^4$

**Michel Planat** [1,*] ✱ ⓘ**, Raymond Aschheim** [2] ⓘ**, Marcelo M. Amaral** [2] ⓘ **and Klee Irwin** [2]

[1]  Institut FEMTO-ST CNRS UMR 6174, Université de Bourgogne/Franche-Comté, 15 B Avenue des Montboucons, F-25044 Besançon, France

[2]  Quantum Gravity Research, Los Angeles, CA 90290, USA; raymond@QuantumGravityResearch.org (R.A.); Marcelo@quantumgravityresearch.org (M.M.A.); Klee@quantumgravityresearch.org (K.I.)

[*]  Correspondence: michel.planat@femto-st.fr

**Abstract:** The authors previously found a model of universal quantum computation by making use of the coset structure of subgroups of a free group $G$ with relations. A valid subgroup $H$ of index $d$ in $G$ leads to a 'magic' state $|\psi\rangle$ in $d$-dimensional Hilbert space that encodes a minimal informationally complete quantum measurement (or MIC), possibly carrying a finite 'contextual' geometry. In the present work, we choose $G$ as the fundamental group $\pi_1(V)$ of an exotic 4-manifold $V$, more precisely a 'small exotic' (space-time) $R^4$ (that is homeomorphic and isometric, but not diffeomorphic to the Euclidean $\mathbb{R}^4$). Our selected example, due to S. Akbulut and R. E. Gompf, has two remarkable properties: (a) it shows the occurrence of standard contextual geometries such as the Fano plane (at index 7), Mermin's pentagram (at index 10), the two-qubit commutation picture $GQ(2,2)$ (at index 15), and the combinatorial Grassmannian $Gr(2,8)$ (at index 28); and (b) it allows the interpretation of MICs measurements as arising from such exotic (space-time) $R^4$s. Our new picture relating a topological quantum computing and exotic space-time is also intended to become an approach of 'quantum gravity'.

**Keywords:** topological quantum computing; 4-manifolds; akbulut cork; exotic $R^4$; fundamental group; finite geometry; Cayley–Dickson algebras

## 1. Introduction

Concepts in quantum computing and the mathematics of 3-manifolds could be combined, as shown in our previous papers [1–4]. In the present work, we consider that quantum computing is performed dynamically in four dimensions interpreted as the Euclidean space-time $R^4$. This offers a preliminary connection to the field called 'quantum gravity' with unconventional methods. In four dimensions, the topological and the smooth structure of a manifold are apart. There exist infinitely many 4-manifolds that are homeomorphic but non-diffeomorphic to each other [5–8]. They can be seen as distinct copies of space-time not identifiable to the ordinary Euclidean space-time.

A cornerstone of our approach is an 'exotic' 4-manifold called an Akbulut cork $W$ that is contractible, compact, and smooth, but not diffeomorphic to the 4-ball [8]. In our approach, we do not need the full toolkit of 4-manifolds since we are focusing on $W$ and its neighbors only. All we need is to understand the handlebody decomposition of a 4-manifold, the fundamental group $\pi_1(\partial W)$ of the three-dimensional boundary $\partial W$ of $W$, and related fundamental groups. Following the methodology of

our previous work, the subgroup structure of such $\pi_1$s corresponds to the Hilbert spaces of interest. Our view is close to the many-worlds interpretation of quantum mechanics where all possible outcomes of quantum measurements are realized in some 'world' and are objectively real [9]. One arrives at a many-manifolds view of quantum computing—reminiscent of the many-worlds—where the many-manifolds are in an exotic class and can be seen as many-quantum generalized measurements, the latter being POVMs (positive operator valued measures).

In quantum information theory, the two-qubit configuration and its properties, namely quantum entanglement and quantum contextuality, have been discussed at length as prototypes of peculiarities or resources in the quantum world. Our model of quantum computing is based on the concept of a magic state—a state that has to be added to the eigenstates of the $d$-dimensional Pauli group—to allow universal quantum computation. This was started by Bravyi and Kitaev in 2005 [10] for qubits ($d = 2$). A subset of magic states consists of states associated to minimal informationally complete measurements, which we called MIC states [1] (see Appendix A for a definition). We require that magic states should be MIC states as well. To obtain the candidate MIC states, one uses the fact that a permutation may be realized as a permutation matrix/gate and that mutually commuting matrices share eigenstates. They are either of the stabilizer type (as elements of the Pauli group) or of the magic type. One keeps magic states that are MIC states in order to preserve complete information during the computation and measurements. The detailed process of quantum computation is not revealed at this stage and will depend on the physical support for the states.

A further step in our quantum computing model is to introduce a three-dimensional manifold $M^3$ whose fundamental group $G = \pi_1(M^3)$ would be the source of MIC states [2,3]. Recall that $G$ is a free group with relations and that a $d$-dimensional MIC state may be obtained from the permutation group that organizes the cosets of an appropriate subgroup of index $d$ of $G$. It was considered by us quite remarkable that two group geometrical axioms very often govern the MIC states of interest [4]: (i) the normal (or conjugate) closure $\{g^{-1}hg|g \in G \text{ and } h \in H\}$ of the appropriate subgroup $H$ of $G$ equals $G$ itself; and (ii) there is no geometry (a triple of cosets does not produce equal pairwise stabilizer subgroups). See [4] (Section 1.1) for our method of building a finite geometry from coset classes and Appendix B for the special case of the Fano plane. However, these rules had to be modified by allowing either the simultaneous falsification of (i) and (ii) or by tolerating a few exceptions. The latter item means a case of geometric contextuality, the parallel to quantum contextuality [11].

We are fortunate that the finite geometrical structures sustaining two-qubit commutation, the projective plane or Fano plane $PG(2,2)$ over the two-element field $F_2$, the generalized quadrangle of order two $GQ(2,2)$, its embedding three-dimensional projective space $PG(3,2)$, and other structures relevant to quantum contextuality, are ingredients in our quantum computing model based on an Akbulut cork. Even more appealing is the recovery of the Grassmannian configuration $Gr(2,8)$ on 28 vertices and 56 lines that connects to Cayley–Dickson algebras up to level 8.

Our description is organized as follows. Section 2 summarizes the peculiarities of small exotic 4-manifolds of type $R^4$. This includes the handlebody structure of an arbitrary 4-manifold in Section 2.1, the concept of an Akbulut cork in Section 2.2, and examples of small exotic 4-manifolds that are homeomorphic but not diffeomorphic to the Euclidean space $\mathbb{R}^4$ in Section 2.3. In Section 3, the connection of quantum computing to a manifold $M$ is through the fundamental group $\pi_1(M)$. The subgroup structure of $\pi_1(M)$ is explored when $M$ is the boundary $\partial W$ of an Akbulut cork (in Section 3.1), the manifold $\bar{W}$ in Akbulut h-cobordism (in Section 3.2), and the middle level $Q$ between diffeomorphic connected sums involving the exotic $R^4$s (in Section 3.3). Section 4 is a discussion of the obtained results and their potential interest for connecting quantum information and the structure of space-time. Calculations in this paper were performed using Magma [12] and SnapPy [13].

## 2. Excerpts on the Theory of 4-Manifolds and Exotic $R^4$s

### 2.1. Handlebody of a 4-Manifold

Let us introduce some excerpts of the theory of 4-manifolds needed for our paper [5–7]. It concerns the decomposition of a 4-manifold into one- and two-dimensional handles, as shown in Figure 1, see [5] (Figure 1.1 and Figure 1.2). Let $B^n$ and $S^n$ be the $n$-dimensional ball and the $n$-dimensional sphere, respectively. An observer is placed at the boundary $\partial B^4 = S^3$ of the 0-handle $B^4$ and watches the attaching regions of the 1- and 2-handles. The attaching region of 1-handle is a pair of balls $B^3$ (the yellow balls), and the attaching region of 2-handles is a framed knot (the red knotted circle) or a knot going over the 1-handle (shown in blue). Notice that the 2-handles are attached after the 1-handles. For closed 4-manifolds, there is no need for visualizing a 3-handle since it can be directly attached to the 0-handle. The 1-handle can also be figured out as a dotted circle $S^1 \times B^3$ obtained by squeezing together the two three-dimensional balls $B^3$ so that they become flat and close together [6] (p. 169), as shown in Figure 1b. For the attaching region of a 2- and a 3-handle, one needs to enrich our knowledge by introducing the concept of an Akbulut cork, as described below. The surgering of a 2-handle to a 1-handle is illustrated in Figure 2a (see also [6] (Figure 5.33)). The 0-framed 2-handle (left) and the 'dotted' 1-handle (right) are diffeomorphic at their boundary $\partial$. The boundary of a 2- and a 3-handle is intimately related to the Akbulut cork shown in Figure 2b, as described in Section 2.3.

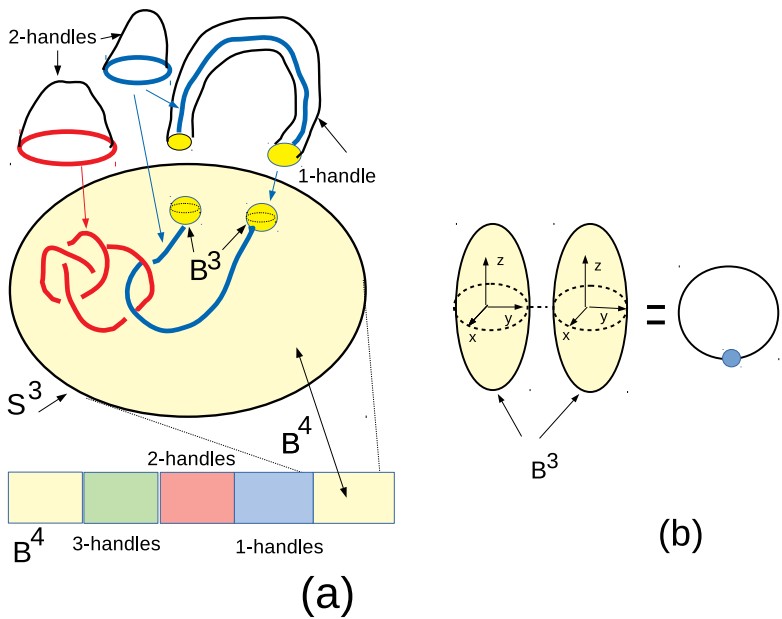

**Figure 1.** (**a**) Handlebody of a 4-manifold with the structure of 1- and 2-handles over the 0-handle $B^4$ ; and (**b**) the structure of a 1-handle as a dotted circle $S^1 \times B^3$.

### 2.2. Akbulut Cork

A Mazur manifold is a contractible, compact, smooth 4-manifold (with boundary) not diffeomorphic to the standard 4-ball $B^4$ [5]. Its boundary is a homology 3-sphere. If one restricts to Mazur manifolds that have a handle decomposition into a single 0-handle, a single 1-handle and a single 2-handle, then the manifold has to be of the form of the dotted circle $S^1 \times B^3$ (as in Figure 2a) (right) union a 2-handle.

Recall that, given $p, q, r$ (with $p \leq q \leq r$), the Brieskorn 3-manifold $\Sigma(p, q, r)$ is the intersection in the complex 3-space $\mathbb{C}^3$ of the five-dimensional sphere $S^5$ with the surface of equation $z_1^p + z_2^q + z_3^r = 0$. The smallest known Mazur manifold is the Akbulut cork $W$ [8,14] pictured in Figure 2b and its boundary is the Brieskorn homology sphere $\Sigma(2, 5, 7)$.

According to Reference [15], there exists an involution $f : \partial W \to \partial W$ that surgers the dotted 1-handle $S^1 \times B^3$ to the 0-framed 2-handle $S^2 \times B^2$ and back, in the interior of $W$. The Akbulut cork is shown in Figure 2b. The Akbulut cork has a simple definition in terms of the framings $\pm 1$ of $(-3, 3, -3)$ pretzel knot also called $K = 9_{46}$ [15] (Figure 3). It has been shown that $\partial W = \Sigma(2, 5, 7) = K(1, 1)$ and $W = K(-1, 1)$.

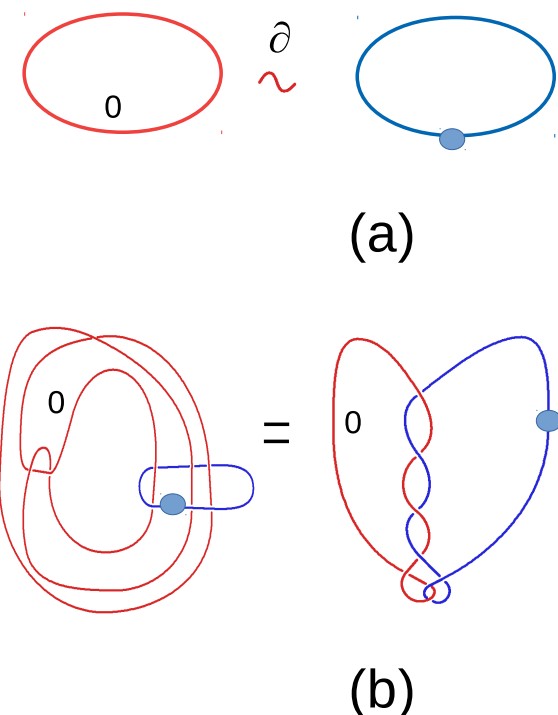

(a)

(b)

**Figure 2.** (**a**) A 0-framed 2-handle $S^2 \times B^2$ (**left**) and a dotted 1-handle $S^1 \times B^3$ (**right**) are diffeomorphic at their boundary $\partial = S^2 \times S^1$; and (**b**) two equivalent pictures of the Akbulut cork $W$.

### *2.3. Exotic Manifold $R^4$*

An exotic $R^4$ is a differentiable manifold that is homeomorphic but not diffeomorphic to the Euclidean space $\mathbb{R}^4$. An exotic $R^4$ is called small if it can be smoothly embedded as an open subset of the standard $\mathbb{R}^4$ and is called large otherwise. Here, we are concerned with an example of a small exotic $R^4$. Let us quote Theorem 1 of [8].

**Theorem 1.** *There is a smooth contractible 4-manifold V with $\partial V = \partial W$, such that V is homeomorphic but not diffeomorphic to W relative to the boundary.*

**Proof.** Sketch of proof [8]:

Let $\alpha$ be a loop in $\partial W$ as in Figure 3a. $\alpha$ is not slice in $W$ (does not bound an imbedded smooth $B^2$ in $W$) but $\phi(\alpha)$ is slice. Then, $\phi$ does not extend to a self-diffeomorphism $\phi : W \to W$.  □

It is time to recall that a cobordism between two oriented $m$-manifolds $M$ and $N$ is any oriented $(m + 1)$-manifold $W_0$ such that the boundary is $\partial W_0 = \bar{M} \cup N$, where $M$ appears with the reverse orientation. The cobordism $M \times [0, 1]$ is called the trivial cobordism. Next, a cobordism $W_0$ between $M$ and $N$ is called an h-cobordism if $W_0$ is homotopically such as the trivial cobordism. The h-cobordism due to S. Smale in 1960 states that, if $M^m$ and $N^m$ are compact simply-connected oriented $M$-manifolds that are h-cobordant through the simply-connected $(m + 1)$-manifold $W_0^{m+1}$, then $M$ and $N$ are diffeomorphic [7] (p. 29). However, this theorem fails in dimension 4. If $M$ and $N$ are cobordant 4-manifolds, then $N$ can be obtained from $M$ by cutting out a compact contractible submanifold $W$

and gluing it back in by using an involution of $\partial W$. The 4-manifold $W$ is a 'fake' version of the 4-ball $B^4$ called an Akbulut cork [7] (Figure 2.23).

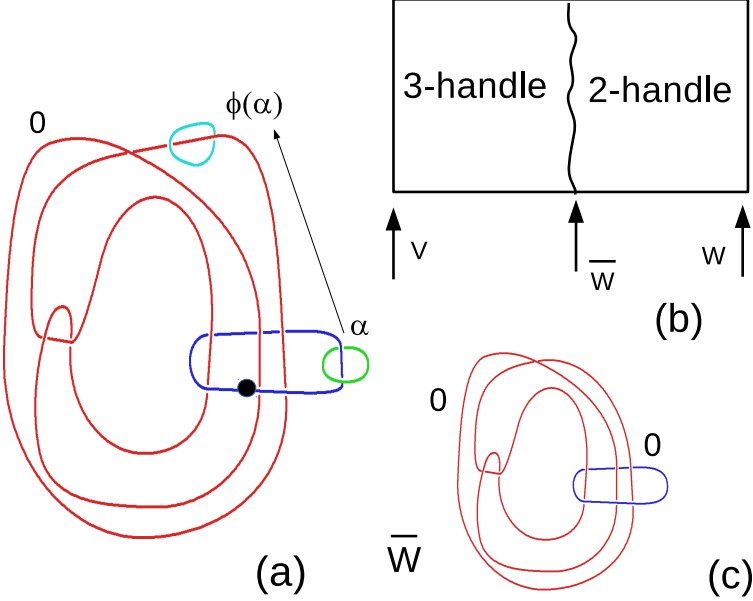

**Figure 3.** (**a**) The loop $\alpha$ is not slice on the Akbulut cork; (**b**) the non-trivial h-cobordism between small exotic manifolds $V$ and $W$; and (**c**) the mediating 4-manifold $\bar{W}$.

The h-cobordism under question in our example may be described by attaching an algebraic cancelling pair of 2- and 3-handles to the interior of Akbulut cork $W$, as pictured in Figure 3b (see [8] (p. 343)). The 4-manifold $\bar{W}$ mediating $V$ and $W$ as shown in Figure 3c [alias the 0-surgery $L7a6(0,1)(0,1)$] (see [8] (p. 355)).

The result is relative since $V$ itself is diffeomorphic to $W$ but such a diffeomorphism cannot extend to the identity map $\partial V \to \partial W$ on the boundary [14]. In the previous reference, two exotic manifolds $Q_1$ and $Q_2$ are built that are homeomorphic but not diffeomorphic to each other in their interior.

By the way, the exotic $R^4$ manifolds $Q_1$ and $Q_2$ are related by a diffeomorphism $Q_1 \# S^2 \times S^2 \approx Q \approx Q_2 \# S^2 \times S^2$ (where # is the connected sum between two manifolds) and $Q$ is called the middle level between such connected sums. This is shown in Figure 4 for the two $R^4$ manifolds $Q_1$ and $Q_2$ [14,16] (Figure 2).

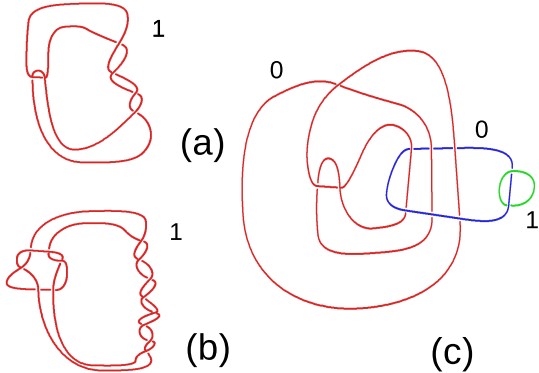

**Figure 4.** Exotic $R^4$ manifolds $Q_1$ shown in (**a**) and $Q_2$ shown in (**b**). The connected sums $Q_1 \# S^2 \times S^2$ and $Q_2 \# S^2 \times S^2$ are diffeomorphic with middle level $Q$ shown in (**c**).

## 3. Finite Geometry of Small Exotic $R^4$s and Quantum Computing

From now, we focus on the relationship between the aforementioned small exotic $R^4$s and a model of quantum computing and (generalized) quantum measurements based on the magic states. The background about the magic states is recalled in the introduction (see also [1–4]). This connection is established by computing the finite index subgroup structure of the fundamental group for the boundary $\partial W$ of Akbulut cork $W$, for $\bar{W}$ in Figure 3 and for $Q$ in Figure 4, and by making use of earlier results of the authors.

In the present paper, we choose $G$ as the fundamental group $\pi_1(M^4)$ of a 4-manifold $M^4$ that is the boundary $\partial W$ of Akbulut cork $W$, or governs the Akbulut h-cobordism. More precisely, one takes the manifold $M^4$ as $\bar{W}$ in Figure 3 and $Q$ in Figure 4. Manifolds $Q_1$ and $Q_2$ are the small exotic $R^4$s introduced in Reference [14] (Figures 1 and 2). There are homeomorphic but not diffeomorphic to each other in their interiors. This choice has two important consequences.

Recall the Introduction that Axioms (i) and (ii) are expected to govern the subgroup structure of groups $G$ relevant to our model of quantum computing based on magic states. For the aforementioned manifolds $M^4$, the fundamental group $G = \pi_1(M^4)$ is such that: (i) it is always satisfied; and (ii) most often it is true or geometric contextuality occurs with corresponding finite geometries of great interest such as the Fano plane $PG(2,2)$ (at index 7), the Mermin's pentagram (at index 10), the finite projective space $PG(3,2)$ or its subgeometry $GQ(2,2)$ that is known to control 2-qubit commutation [4] (Figure 1) (at index 15), the Grassmannian Gr$(2,8)$ containing Cayley–Dickson algebras (at index 28), and a few maximally multipartite graphs.

Second, this new frame provides a physical interpretation of quantum computation and measurements as follows. Let us imagine that $\mathbb{R}^4$ is our familiar space-time. Thus, the 'fake' 4-ball $W$—the Akbulut cork—allows the existence of smoothly embedded open subsets of space-time—the exotic $R^4$ manifolds such as $Q_1$ and $Q_2$—that we interpret in this model as 4-manifolds associated to quantum measurements.

### 3.1. The Boundary $\partial W$ of Akbulut Cork

As announced above, $\partial W = K(1,1) \equiv \Sigma(2,5,7)$ is a Brieskorn sphere with fundamental group

$$\pi_1(\Sigma(2,5,7)) = \left\langle a, b | aBab^2aBab^3, a^4bAb \right\rangle, \text{where } A = a^{-1}, B = b^{-1}.$$

The cardinality structure of subgroups of this fundamental group is found to be the sequence

$$\eta_d[\pi_1(\Sigma(2,5,7))] = [0,0,0,0,0,0,2,1,0,3,0,0,0,\mathbf{12},\mathbf{145},\mathbf{178},47,0,0,\mathbf{4},\cdots].$$

All the subgroups $H$ of the above list satisfy Axiom (i).

Up to index 28, exceptions to Axiom (ii) can be found at index $d = 14, 16, 20$ featuring the geometry of multipartite graphs $K_2^{(d/2)}$ with $d/2$ parties, at index $d = 15$, and finally at index 28. Here and below, the bold notation features the existence of such exceptions.

Apart from these exceptions, the permutation group organizing the cosets is an alternating group $A_d$. The coset graph is the complete graph $K_d$ on $d$ vertices. One cannot find a triple of cosets with strictly equal pairwise stabilizer subgroups of $A_d$ (no geometry), thus (ii) is satisfied.

At index 15, when (ii) is not satisfied, the permutation group organizing the cosets is isomorphic to $A_7$. The stabilized geometry is the finite projective space $PG(3,2)$ (with 15 points, 15 planes, and 35 lines), as illustrated in Figure 5a. A different enlightening of the projective space $PG(3,2)$ appears in [17] with the connection to the Kirkman schoolgirl problem. The geometry is contextual in the sense that all lines not going through the identity element do not show mutually commuting cosets.

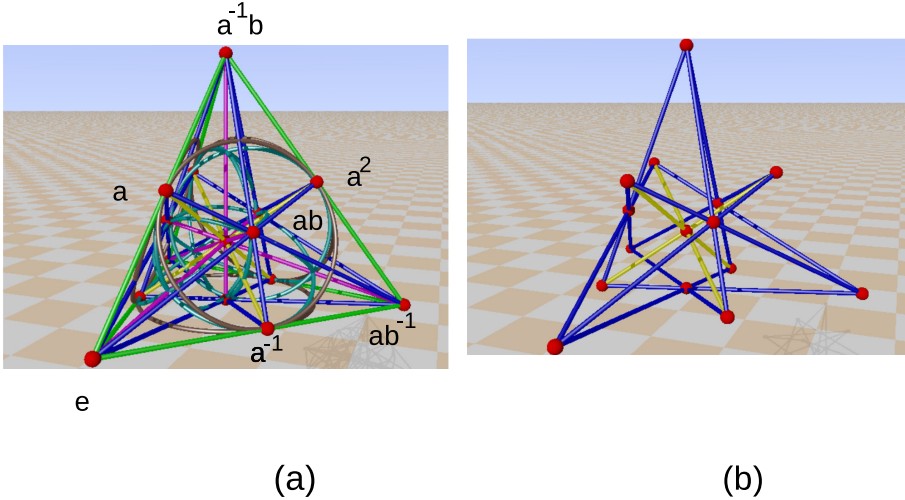

**Figure 5.** (**a**) A picture of the smallest finite projective space $PG(3,2)$. It is found at Frans Marcelis'
website [18]. The coset coordinates shown on the Fano plane with red bullets of $PG(3,2)$ correspond
the case in Table 2. (**b**) A picture of the generalized quadrangle of order two $GQ(2,2)$ embedded in
$PG(3,2)$. It may also be found at Frans Marcelis' website.

At index 28, when (ii) is not satisfied, there are two cases. In the first case, the group $P$ is of order
$2^8 8!$ and the geometry is the multipartite graph $K_4^{(7)}$. In the second case, the permutation group is
$P = A_8$ and the geometry is the configuration $[28_6, 56_3]$ on 28 points and 56 lines of size 3. It has been
shown that the geometry in question corresponds to the combinatorial Grassmannian of type $\mathrm{Gr}(2,8)$,
alias the configuration obtained from the points off the hyperbolic quadric $Q^+(5,2)$ in the complex
projective space $PG(5,2)$ [19]. Interestingly, $\mathrm{Gr}(2,8)$ can be nested by gradual removal of a so-called
'Conwell heptad' and be identified to the tail of the sequence of Cayley–Dickson algebras [19,20]
(Table 4).

One expects a connection of the 28-point configuration to a del Pezzo surface of degree 2 (since
the 56 lines of such a del Pezzo surface map in pairs to the 28 bitangents of a quartic). With the
Freudenthal–Tits magic square, there should also exist a link of such a symmetry to octooctonions and
the Lie group $E_8$ [21,22] that may be useful for particle physics. Further, a Cayley–Dickson sequence is
proposed to connect quantum information and quantum gravity in [23].

The $[28_6, 56_3]$ configuration on 28 points and 56 lines.

Below are given some hints about the configuration that is stabilized at the index 28 subgroup $H$
of the fundamental group $\pi_1(\partial W)$ whose permutation group $P$ organizing the cosets is isomorphic to
$A_8$. Recall that $\partial W$ is the boundary of Akbulut cork $W$. The 28-letter permutation group $P$ has two
generators as follows

$$P = \langle 28 | g_1, g_2 \rangle \ with g_1 = (2,4,8,6,3)(5,10,15,13,9)(11,12,18,25,17)$$
$$(14,20,19,24,21)(16,22,26,28,23), g_2 = (1,2,5,11,6,7,3)(4,8,12,19,22,14,9)$$
$$(10,16,24,27,21,26,17)(13,20,18,25,28,23,15).$$

Using the method described in Appendix B, one derives the configuration $[28_6, 56_3]$ on 28
points and 56 lines. As shown in [19] (Table 4), the configuration is isomorphic to the combinatorial
Grassmannian $\mathrm{Gr}(2,8)$ and nested by a sequence of binomial configurations isomorphic to $\mathrm{Gr}(2,i)$,
$i \leq 8$, associated with Cayley–Dickson algebras. This statement is checked by listing the 56 lines on
the 28 points of the configuration as follows

$$\{1,7,27\}, \rightarrow \mathbf{Gr(2,3)}$$
$$\{1,15,23\}, \{15,17,27\}, \{7,17,23\}, \rightarrow \mathbf{Gr(2,4)}$$
$$\{1,5,26\}, \{5,18,27\}, \{5,15,24\}, \{23,24,26\}, \{17,18,24\}, \{7,18,26\}, \rightarrow \mathbf{Gr(2,5)}$$
$$\{12,14,17\}, \{1,9,22\}, \{5,8,9\}, \{9,14,15\}, \{7,12,22\}, \{8,12,18\},$$
$$\{8,14,24\}, \{8,22,26\}, \{14,22,23\}, \{9,12,27\}, \rightarrow \mathbf{Gr(2,6)}$$
$$\{3,10,15\}, \{3,6,24\}, \{3,17,25\}, \{3,23,28\}, \{1,10,28\}, \{3,14,19\}, \{7,25,28\}, \{6,8,19\},$$
$$\{19,22,28\}, \{5,6,10\}, \{12,19,25\}, \{10,25,27\}, \{9,10,19\}, \{6,18,25\}, \{6,26,28\}, \rightarrow \mathbf{Gr(2,7)}$$
$$\{4,11,12\}, \{11,21,25\}, \{6,20,21\}, \{2,3,21\}, \{2,4,14\}, \{7,11,16\}, \{2,16,23\}, \{1,13,16\},$$
$$\{2,11,17\}, \{4,19,21\}, \{16,20,26\}, \{2,13,15\}, \{11,13,27\}, \{16,21,28\}, \{2,20,24\},$$
$$\{5,13,20\}, \{11,18,20\}, \{4,9,13\}, \{4,8,20\}, \{4,16,22\}, \{10,13,21\} \rightarrow \mathbf{Gr(2,8)}.$$

More precisely, the distinguished configuration $[21_5, 35_3]$ isomorphic to $\mathrm{Gr}(2,7)$ in the list above is stabilized thanks to the subgroup of $P$ isomorphic to $A_7$. The distinguished Cayley–Salmon configuration $[15_4, 20_3]$ isomorphic to $\mathrm{Gr}(2,6)$ in the list is obtained thanks to one of the two subgroups of $P$ isomorphic to $A_6$. The upper stages of the list correspond to a Desargues configuration $[10_3, 10_3]$, to a Pasch configuration $[6_2, 4_3]$, and to a single line $[3_1, 1_3]$ and are isomorphic to the Grassmannians $\mathrm{Gr}(2,5)$, $\mathrm{Gr}(2,4)$, and $\mathrm{Gr}(2,3)$, respectively. The Cayley–Salmon configuration configuration is shown in Figure 6, see also [20] (Figure 12). For the embedding of Cayley–Salmon configuration into $[21_5, 35_3]$ configuration, see [20] (Figure 18).

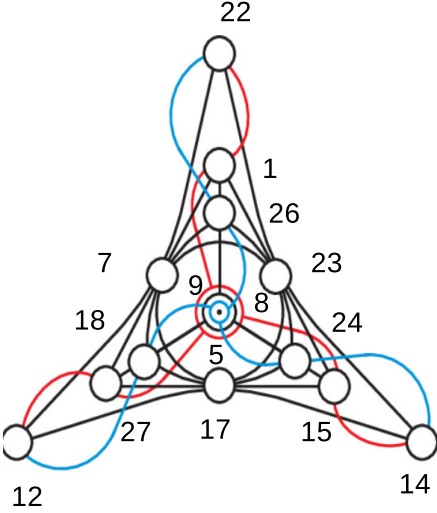

**Figure 6.** The Cayley–Salmon configuration built around the Desargues configuration (itself built around the Pasch configuration) as in [20] (Figure 12).

Frank Marcelis provided a parameterization of the Cayley–Salmon configuration in terms of 3-qubit operators [18].

Not surprisingly, geometric contextuality (in the coset coordinatization not given here) is a common feature of all lines except for the ones going through the identity element.

As a final note for this subsection, we find Brieskorn spheres other than $\Sigma(2,5,7)$ whose fundamental group admits an index 28 subgroup isomorphic to $A_8$ whose geometry is the configuration with 28 points and 56 lines. Three-manifolds $\Sigma(2,4,7)$, $\Sigma(3,4,5)$, $\Sigma(3,4,7)$, and $\Sigma(3,5,7)$ are such Brieskorn spheres.

### 3.2. The Manifold $\bar{W}$ Mediating the Akbulut Cobordism between Exotic Manifolds V and W

The fundamental group $\pi_1(\bar{W})$ of the h-cobordism $\bar{W}$ is as follows

$$\pi_1(\bar{W}) = \left\langle a, b | a^3 b^2 A B^3 A b^2, (ab)^2 a B^2 A b^2 A B^2 \right\rangle.$$

The cardinality structure of subgroups of this fundamental group is

$$\eta_d[\pi_1(\bar{W})] = [0, 0, 0, 0, 1, 1, 2, 0, 0, \mathbf{1}, 0, \mathbf{5}, \mathbf{4}, \mathbf{9}, \mathbf{7}, 1 \cdots]$$

As is the previous subsection, all subgroups $H$ of the above list satisfy Axiom (i). When Axiom (ii) is not satisfied, the geometry is contextual. All cases are listed in Table 1. Columns 4 and 5 give details about the existence of a MIC at the corresponding dimension when this can be checked, i.e., the cardinality of $P$ is low enough.

**Table 1.** Geometric structure of subgroups of the fundamental group $\pi_1(\bar{W})$ for the h-cobordism $\bar{W}$ between exotic manifolds $V$ and $W$. Bold characters are for the contextual geometries. $G_{1092}$ is the simple group of order 1092. When a MIC state can be found, details are given in Column 4, while the number $pp$ of distinct values of pairwise products in the MIC is in Column 5.

| d | P | Geometry | MIC Fiducial | pp |
|---|---|---|---|---|
| 5 | $A_5$ | $K_5$ | $(0, 1, -1, -1, 1)$ | 1 |
| 6 | $A_6$ | $K_6$ | $(1, \omega_6 - 1, 0, 0, -\omega_6, 0)$ | 2 |
| 7 | $A_7$ | $K_7$ | | |
| 10 | $A_5$ | **Mermin pentagram** | no | |
| 12 | $A_{12}$ | $K_{12}$ | | |
| | $A_5$ | **K(2,2,2,2,2,2)** | no | |
| 13 | $A_{13}$ | $K_{13}$ | | |
| 14 | $G_{1092}, A_{14}$ | $K_{14}$ | | |
| | $2^6 \rtimes A_5$ | **K(2,2,2,2,2,2,2)** | | |
| 15 | $A_{15}$ | $K_{15}$ | | |
| | $A_5$ | **K(3,3,3,3,3)** | yes | 3 |
| | $A_7$ | **PG(3,2)** | | |
| 16 | $A_{16}$ | $K_{16}$ | | |

A picture of (contextual) Mermin pentagram stabilized at index 10 is given in Figure 7.

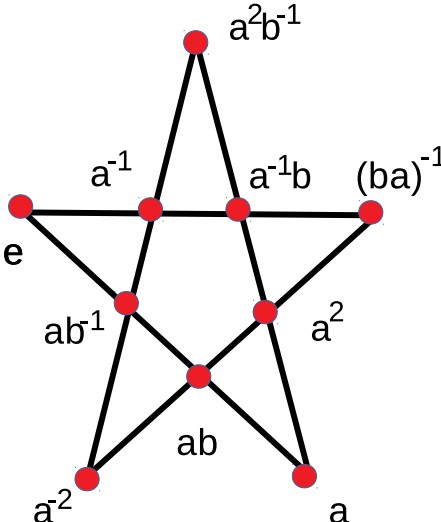

**Figure 7.** Mermin pentagram as the contextual geometry occuring for the index 10 subgroup in the fundamental group $\pi_1(\bar{W})$ of the h-cobordism $\bar{W}$ between exotic manifolds $V$ and $W$.

### 3.3. The Middle Level Q between the Diffeomorphic Connected Sums

The fundamental group for the middle level $Q$ between diffeomorphic connected sums $Q_1 \# S^2 \times S^2$ and $Q_2 \# S^2 \times S^2$ is as follows

$$\pi_1(Q) = \left\langle a, b \,|\, a^2 b A^2 b a^2 B A B, a^2 B A^3 B a^2 b^3 \right\rangle.$$

The cardinality structure of subgroups of this fundamental group is

$$\eta_d[\pi_1(Q)] = [0, 0, 0, 0, 0, 2, \mathbf{2}, 1, 0, 1, 0, 0, 0, \mathbf{2}, \mathbf{2}, 3 \cdots]$$

As in the previous subsection, all the subgroups $H$ of the above list satisfy Axiom (i). When Axiom (ii) is not satisfied, the geometry is contextual. All cases are listed in Table 2. The coset coordinates of the (contextual) Fano plane are shown on Figure 5a (see also Figure A1 in Appendix B). Columns 4 and 5 give details about the existence of a MIC at the corresponding dimension when this can be checked, i.e., if the cardinality of $P$ is low enough. Notice that, at index 15, both the the finite projective space $PG(3,2)$ and the embedded generalized quadrangle of order two $GQ(2,2)$ (shown in Figure 5) are stabilized. Both geometries are contextual. The latter case is pictured in [4] (Figure 1) with a double coordinatization: the cosets or the two-qubit operators.

These results about the h-cobordism of $R^4$ exotic manifolds and the relation to quantum computing, as developed in our model, are encouraging us to think about a physical implementation. However, this is left open in this paper.

**Table 2.** Geometric structure of subgroups of the fundamental group $\pi_1(Q)$ for the middle level $Q$ of Akbulut's *h*-cobordism between connected sums $Q_1 \# S^2 \times S^2$ and $Q_2 \# S^2 \times S^2$. Bold characters are for the contextual geometries. When a MIC state can be found, details are given in Column 4, while the number $pp$ of distinct values of pairwise products in the MIC is in Column 5. $GQ(2,2)$ is the generalized quadrangle of order two embedded in PG(3,2), as shown in Figure 5b.

| d | P | Geometry | MIC Fiducial | pp |
|---|---|---|---|---|
| 6 | $A_6$ | $K_6$ | $(1, \omega_6 - 1, 0, 0, -\omega_6, 0)$ | 2 |
| 7 | $PSL(2,7)$ | **Fano plane** | $(1, 1, 0, -1, 0, -1, 0)$ | 2 |
| 8 | $PSL(2,7)$ | $K_8$ | no | |
| 10 | $A_6$ | $K_{10}$ | yes | 5 |
| 14 | $PSL(2,7)$ | **K(2,2,2,2,2,2,2)** | no | |
| 15 | $A_6$ | **PG(3,2), GQ(2,2)** | yes | 4 |
| 16 | $A_{16}$ | $K_{16}$ | no | |
| | $SL(2,7)$ | K(2,2,2,2,2,2,2,2) | no | |

## 4. Conclusions

As recalled in Section 2.3, the h-cobordism theorem in four dimensions is true topologically—there is a homotopy equivalence of the inclusion maps of four-dimensional manifolds $M$ and $N$ into the five-dimensional cobordism—but is false piecewise linearly and smoothly—a piecewise linear structure or atlas cannot be smoothed in an unique way. These statements had to await topologists Michael Friedmann and Simon Donaldson in the twentieth century to be rigorously established [5–7]. Handle decomposition in a way described in Section 2, handle trading, handle cancellation, and handle sliding are the main techniques for establishing the failure of the four-dimensional h-cobordism theorem and the possible existence of infinitely many (exotic) $R^4$ manifolds that are homeomorphic but not diffeomorphic to the Euclidean $\mathbb{R}^4$. It has been emphasized that the non-trivial h-cobordism between exotic $R^4$ manifolds may occur through a 'pasting' ingredient called an Akbulut cork $W$ whose three-dimensional boundary $\partial W$ is described explicitly as the Brieskorn sphere $\Sigma(2,5,7)$.

We use the fundamental group $\pi_1(\partial W)$ as a witness of symmetries encoded into the 4-manifolds under investigation. Interpreting $\mathbb{R}^4$ as the ordinary space-time, the symmetries have potential applications to space-time physics (e.g., cosmology) and quantum mechanics (e.g., particle physics

and quantum computation). Attempts to connect the issues of geometry and topology of exotic $R^4$ to quantum gravity have been developed in other papers (e.g., [24,25]). Another promising approach is in [26]. In the past, the Brieskorn sphere $\Sigma(2,3,5)$, alias the Poincaré dodecahedral space, was proposed as the 'shape' of space because such a 3-manifold—the boundary of the 4-manifold $E_8$—explains some anomalies in the cosmic microwave background [27,28]. It would be interesting to use $\Sigma(2,5,7)$ as an alternative model in the line of work performed in [29,30] and related work.

We find in Section 3 subgroups of index 15 and 28 of $\pi_1(\partial W)$ connecting to the three-dimensional projective space $PG(3,2)$ and to the Grassmannian $Gr(2,8)$, respectively. The former finite geometry has relevance to the two-qubit model of quantum computing and the latter geometry connect to Cayley–Dickson algebras already used for particle physics. In our previous work, we found that MICs (minimal informationally complete POVMs) built from finite index subgroups of the fundamental group $\pi_1(M^3)$ of a 3-manifold $M^3$ serve as models of universal quantum computing (UQC). A MIC-UQC model based on a finite index subgroup of the fundamental group $\pi_1(M^4)$ of an exotic $M^4$ has interest for interpreting (generalized) quantum measurements as corresponding to the non-diffeomorphic branches (to the exotic copies) of $M^4$ or its submanifolds.

To conclude, we are confident that an exotic 4-manifold approach of theoretical physics allows establishing bridges between space-time physics and quantum physics as an alternative or as a complement to string theory and loop quantum gravity.

**Author Contributions:** All authors contributed significantly to the conceptualization and methodology of the paper. M.P. wrote the paper and coauthors R.A., M.M.A., and K.I. gave final corrections and approval for publication. All authors have read and agreed to the published version of the manuscript.

**Funding:** Funding was obtained from Quantum Gravity Research in Los Angeles, CA.

**Conflicts of Interest:** The author declares no competing interests.

## Appendix A

A POVM is a collection of positive semi-definite operators $\{E_1, \ldots, E_m\}$ that sum to the identity. In the measurement of a state $\rho$, the $i$th outcome is obtained with a probability given by the Born rule $p(i) = \text{tr}(\rho E_i)$. For a minimal and informationally complete POVM (or MIC), one needs $d^2$ one-dimensional projectors $\Pi_i = |\psi_i\rangle \langle \psi_i|$, with $\Pi_i = dE_i$, such that the rank of the Gram matrix with elements $\text{tr}(\Pi_i \Pi_j)$, is precisely $d^2$.

A SIC (a symmetric informationally complete POVM) obeys the remarkable relation

$$\left| \langle \psi_i | \psi_j \rangle \right|^2 = \text{tr}(\Pi_i \Pi_j) = \frac{d\delta_{ij} + 1}{d + 1},$$

that allows the recovery of the density matrix as

$$\rho = \sum_{i=1}^{d^2} \left[ (d+1)p(i) - \frac{1}{d} \right] \Pi_i.$$

This type of quantum tomography is often known as quantum-Bayesian, where the $p(i)$s represent agents' Bayesian degrees of belief, because the measurement depends on the filtering of $\rho$ by the selected SIC (for an unknown classical signal, this looks similar to the frequency spectrum) [31].

New MICs have been derived with Hermitian angles $\left| \langle \psi_i | \psi_j \rangle \right||_{i \neq j} \in A = \{a_1, \ldots, a_l\}$, a discrete set of values of small cardinality $l$ [1]. A SIC is equiangular with $|A| = 1$ and $a_1 = \frac{1}{\sqrt{d+1}}$.

## Appendix B

Let us summarize how a finite geometry is built from coset classes [4,11].

Let $H$ be a subgroup of index $d$ of a free group $G$ with generators and relations. A coset table over the subgroup $H$ is built by means of a Coxeter–Todd algorithm. Given the coset table, one builds a permutation group $P$ that is the image of $G$ given by its action on the cosets of $H$.

First, one asks that the $d$-letter group $P$ act faithfully and transitively on the set $\Omega = \{1, 2, \cdots, d\}$. The action of $P$ on a pair of distinct elements of $\Omega$ is defined as $(\alpha, \beta)^p = (\alpha^p, \beta^p)$, $p \in P$, $\alpha \neq \beta$. The number of orbits on the product set $\Omega \times \Omega$ is called the rank $r$ of $P$ on $\Omega$. Such a rank of $P$ is at least two, and it also known that two-transitive groups may be identified to rank two permutation groups.

One selects a pair $(\alpha, \beta) \in \Omega \times \Omega$, $\alpha \neq \beta$ and one introduces the two-point stabilizer subgroup $P_{(\alpha, \beta)} = \{p \in P | (\alpha, \beta)^p = (\alpha, \beta)\}$. There are $1 < m \leq r$ such non-isomorphic (two-point stabilizer) subgroups of $P$. Selecting one of them with $\alpha \neq \beta$, one defines a point/line incidence geometry $\mathcal{G}$ whose points are the elements of the set $\Omega$ and whose lines are defined by the subsets of $\Omega$ that share the same two-point stabilizer subgroup. Two lines of $\mathcal{G}$ are distinguished by their (isomorphic) stabilizers acting on distinct subsets of $\Omega$. A non-trivial geometry is obtained from $P$ as soon as the rank of the representation $\mathcal{P}$ of $P$ is $r > 2$, and, at the same time, the number of non isomorphic two-point stabilizers of $\mathcal{P}$ is $m > 2$. Further, $\mathcal{G}$ is said to be *contextual* (shows *geometrical contextuality*) if at least one of its lines/edges is such that a set/pair of vertices is encoded by non-commuting cosets [11].

One of the simplest geometries obtained from coset classes is that of the Fano plane shown in Figure A1 (see also Figures 5–7).

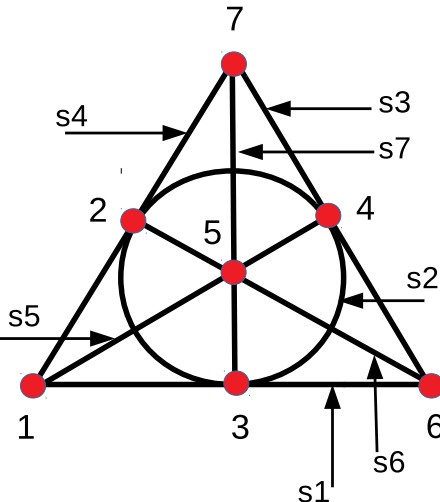

**Figure A1.** The Fano plane as the geometry of the subgroup of index 7 of fundamental group $\pi_1(\bar{W})$ for the manifold $\bar{W}$ mediating the Akbulut cobordism between exotic manifold V and W. The cosets of $\pi_1$ are organized through the permutation group $P = \langle 7 | (1, 2, 4, 5, 6, 7, 3), (2, 5, 6)(3, 7, 4) \rangle$. The cosets are labeled as $[1, \ldots, 7] = [e, a, a^{-1}, a^2, ab, ab^{-1}, a^{-1}b]$. The two-point stabilizer subgroups of $P$ for each line are distinct (acting on different subsets) but isomorphic to each other and to the Klein group $\mathbb{Z}_2 \times \mathbb{Z}_2$. They are as follows: $s_1 = \langle (2, 7)(4, 5), (2, 4)(5, 7) \rangle$, $s_2 = \langle (1, 6)(5, 7), (1, 7)(5, 6) \rangle$, $s_3 = \langle (1, 5)(2, 3), (1, 2)(3, 5) \rangle$, $s_4 = \langle (3, 5)(4, 6), (3, 6)(4, 5) \rangle$, $s_5 = \langle (2, 6)(3, 7), (2, 7)(3, 6) \rangle$, $s_6 = \langle (1, 7)(3, 4), (1, 4)(3, 7) \rangle$, and $s_7 = \langle (1, 2)(4, 6), (1, 6)(2, 4) \rangle$.

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

**Sample Availability:** The codes were written on softwares Snappy and Magma. Part of the calculation is detailed in Section 3.1 and Appendix B.

