# Peer review of "Quantum Computation and Measurements from an Exotic Space-Time R4"

_symmetry, doi:10.3390/sym12050736_

Round 1

Reviewer 1 Report

This paper is a very interesting exposition of properties of differentiable structures in four manifolds. If it were billed as such an exposition it would be an acceptable paper. But the paper proposes to explain how to do quantum computation and measurements using an exotic space time four-manifold. The connections with quantum computation and measurement are nowhere made specific. I recommend rejection of the paper.

Author Response

"This paper is a very interesting exposition of properties of differentiable structures in four manifolds. If it were billed as such an exposition it would be an acceptable paper. But the paper proposes to explain how to do quantum computation and measurements using an exotic space time four-manifold. The connections with quantum computation and measurement are nowhere made specific.

I recommend rejection of the paper."

At least we are happy that our exposition is interesting and mathematically correct.The relation to quantum computing was however explained in the text as part of Section 3. To feature this relation we forwarded these explanations in the introduction. Hopefully the link of our approach based on magic states and informationally complete POVMS (MICs) is becoming more transparent.

Reviewer 2 Report

The authors propose a model for embedding a kind of universal quantum computing in four-dimensional space-time.

Their work relies on the theory of four-manifolds and more specifically on exotic four-manifolds which may be homeomorphic but not diffeomorphic to each other. The fundamental group π1 associated to the description of such exotic manifolds admits subgroups of finite index that are used by the authors as possible models of quantum computing with complete information. In section II, the authors introduce basic objects of geometric topology needed for their work, in particular the concept of an Akbulut cork and the failure of h-cobordism theorem in four dimensions. In section III, they show how the relevant π1 is used for their models of quantum computing and the relation to finite geometries. At index 28, they find a surprising connection to Cayley-Dickson algebras through a 28-point and 56-line configuration singled out in Saniga’s paper.

The work is free of mathematical mistakes.  It is quite interesting that a deep piece of mathematics, i.e., exotic four-manifolds, can be related to the fundamental problem of quantum measurements.

Besides, the connection of the subgroups of π1(∂W) to projective geometries is another interesting issue of the work.

I would like to read subsequent papers of the authors for a possible connection of the 28-point configuration to a del Pezzo surface of degree 2 (since the 56 lines of such a del Pezzo surface map in pairs to the 28 biangents of a quartic).

I have a few comments.

  1. Is the Brieskorn sphere ∂(W) = Σ(2,5,7) in section III A the only one which permits the occurence of the 28-point and 56-line configuration?
  2. Is Akbulut’s cork related to an Alexander horned sphere?
  3. I know about a similar paper ‘Quantum computing in four spatial dimensions’ https://www.preprints.org/manuscript/201905.0021/v1 possibly related to the present paper.
  4. May be the connection to a del Pezzo surface of degree 2 is worthwhile to be mentioned.
  5. A good reference to the many-world interpretation mentioned by the authors in the introduction is lacking. I recommend the publication of this paper after minor changes. 

Author Response

We thank referee two for his review of our manuscript. The minor remarks he wrote as addressed as follows.

"1. Is the Brieskorn sphere ∂(W) = Σ(2,5,7) in section 3.1 the only one which permits the occurence of the 28-point and 56-line configuration?"

The referee is right, there are other Brieskorn spheres found to reveal such a configuration as added at the end of Sec. 3.1.

"2. Is Akbulut’s cork related to an Alexander horned sphere?"

We did not introduce the Alexander horned sphere that corresponds to a wild embedding in 3-dimensional Euclidean space. It is a nice object not directly related to Brieskorn spheres.

" 3. I know about a similar paper ‘Quantum computing in four spatial dimensions’ https://www.preprints.org/manuscript/201905.0021/v1 possibly related to the present paper."

This paper is relevant to our work and we quoted it.

" 4. May be the connection to a del Pezzo surface of degree 2 is worthwhile to be mentioned."

The referee is right. We mentioned this correspondance in Sec. 3. The consequences should be examined further in a future work.

"5. A good reference to the many-world interpretation mentioned by the authors in the introduction is lacking. "

We quoted a general reference by B.~S. de Witt in Physics Today about this topic.

Reviewer 3 Report

The paper deals with quantum computing realised within the framework of smoothness topologies/geometries in dimension 4 where open smooth manifolds can carry infinitely continuum many different nondiffeomorphic smoothness structures. The particular interest of the Authors is in small exotic R4 which are homeomorphic to the standard R4 and are nondiffeomorphic to it. They all are embeddable in the 4-sphere S4 and are derivable from the failure of the h-cobordism theorem in dim 5. In particular there exists the Akbulut cork A in h-cobordant compact 4-manifolds V1 and V2 which is compact contractible 4-manifold with boundary the Brieskorn homology 3-sphere \Sigma(2,5,7). As shown by Authors this A represents elements of quantum computing developed in other papers also by Authors. In particular there are two axioms (i) and (ii) directly referring to quantum computing (and entanglement in particular). The fundamental group of the boundary of A is then shown as respecting (i) and geometrical exceptions of (ii). However, this fulfilment is not that exceptional, it is rather generic (e.g. (i) is usually fulfilled).

I think the paper consists of important and innovative contribution to the field of quantum computing and 4-manifolds and it should be published in Symmetry. However, prior to publication Authors are asked to address three issues below which would make their paper more accessible and more complete.

1. The paper is about quantum computing but specific information about that field are expelled to external sources and works. For the convenience and interest of a general reader it would be highly desirable to explain some motivations behind quantum computing as realised specifically by geometric means of e.g. universal states and others. Authors just mentioned that entanglement and contextuality are vital in quantum computing and that they are reflected in finite geometries in this boundary A context. I would suggest to add just few sentences explaining and justifying the way from finite geometries to quantum contextuality and entanglement such that the name quantum computing in this specific context would become more evident. Even though the problems are widely discussed and presented elsewhere they can be briefly recapitulated here.

2. After addressing 1. above the Authors could briefly explain quantum computation on exotic R4. Namely what is specific to quantum computation which comes from exotic smoothness of R4? In particular, the boundary of A serves as one of the possible whole class of realizations of MIC states and universal quantum computation, so this is highly non-unique and MIC states can be realized in many other contexts. What is specific and unique for quantum computing on exotic R4 which distinguishes that case? One indication is that exotic R4’s are carriers of QM and the factor III_1 of von Neuman algebras and QG. The point 3. below develops farther that issue.

3. Related to quantum computing realized by geometry and topology of (submanifolds) of exotic R4 there naturally appears issue of quantum gravity (QG). However, there exist several publications from early 2000 to the present, investigating the relation of exotic R4’s quantum mechanics and QG. This fact should be somehow reflected in the paper because it seems plausible that QM on exotic R4 could substantiate quantum computations on exotic R4 as in 2. above, i.e. both are not entirely independent.

Below there are two examples (from quite a numerous possible) of publications in the field.

1. T. Asselmeyer-Maluga, J. Król, K. Bielas, P. Klimasara,  From Quantum to Cosmological Regime. The Role of Forcing  and Exotic 4-Smoothness, Universe, 3(2), 31 (2017).

2. G. Etesi, The von Neumann algebra of smooth four-manifolds and a quantum theory of space-time and gravity, arXiv:1712.01828

Missprints

- 1st sentence: something missing, possibly “Concepts in quantum computing and the mathematics of 3-manifolds could be combined in our previous papers” → “Concepts in quantum computing and the mathematics of 3-manifolds could be combined as shown in our previous papers”

- 2nd sentence: it seems too vague “… four dimensions are interpreted as space-time...” (what kind of 4-manifolds?)

- p.12 last sentence: “But this left open in this paper” → “But this is left open in this paper”

Author Response

We thank referee 3 for his detailed reading of our manuscript. We accounted for his main remarks as follows.

1." The paper is about quantum computing but specific information about that field are expelled to external sources and works. For the convenience and interest of a general reader it would be highly desirable to explain some motivations behind quantum computing as realised specifically by geometric means of e.g. universal states and others. Authors just mentioned that entanglement and contextuality are vital in quantum computing
and that they are reflected in finite geometries in this boundary A context. I would suggest to add just few sentences explaining and justifying the way from finite geometries to quantum contextuality and entanglement such that the name quantum computing in this specific context would become more evident. Even though the problems are widely discussed and presented elsewhere they can be briefly recapitulated here."

The relation to quantum computing in our previous work was explained in the text as part of Section 3. To feature this relation we forwarded these explanations in the introduction. Hopefully the link of our approach based on magic states and informationally complete POVMS (MICs) is becoming more transparent.

2. "After addressing 1. above the Authors could briefly explain quantum computation on exotic R4. Namely what is specific to quantum computation which comes from exotic smoothness of R4? In particular, the boundary of A serves as one of the possible whole class of realizations of MIC states and universal quantum computation, so this is highly non-unique and MIC states can be realized in many other contexts. What is specific and unique
for quantum computing on exotic R4 which distinguishes that case? One indication is that exotic R4’s are carriers of QM and the factor III_1 of von Neuman algebras and QG. The point 3. below develops farther that issue."

We think that the improved introduction addresses this point. In particular we wrote:
Our view is close to the many-worlds interpretation of quantum mechanics where all possible outcomes of quantum measurements are realized in some 'world' and are objectively real \cite{DeWitt1970}. One arrives at a many-manifolds view of quantum computing

-reminiscent of the many-worlds- where the many-manifolds are in an exotic class and can be seen as many-quantum generalized measurements, the latter being POVM's (positive operator valued measures).

3. "Related to quantum computing realized by geometry and topology of (submanifolds) of exotic R4 there naturally appears issue of quantum gravity (QG).

However, there exist several publications from early 2000 to the present, investigating the relation of exotic R4’s quantum mechanics and QG. This fact should be somehow reflected in the paper because it seems plausible that QM on exotic R4 could substantiate quantum computations on exotic R4 as in 2. above, i.e. both are not entirely independent.

Below there are two examples (from quite a numerous possible) of publications in the field."

We added these important references in the conclusion. Further work would ne neccessary to digest these concepts and relate them to the present work.

At this stage we are unable to perform tis task.

"Misprints"

The misprints have been corrected.

Round 2

Reviewer 1 Report

It is still the case that the relationship with quantum computing is not adequately explained. I recommend rejection of the paper.